# How Does Destination Experience Value Affect Brand Value and Behavioral Intention? The Moderator Role of Self Congruity

Ebru Gozen [1], Aylin Aktas Alan [2,*], Emel Celep [3], Gozde Seval Ergun [4], Ozgur Yayla [1,*], Huseyin Keles [2] and Arif Aytekin [5]

1   Department of Recreation Management, Manavgat Tourism Faculty, Akdeniz University, Antalya 07600, Türkiye; ebrugozen@akdeniz.edu.tr
2   Department of Tourism Guidance, Manavgat Tourism Faculty, Akdeniz University, Antalya 07600, Türkiye; hkeles@akdeniz.edu.tr
3   Department of Business, Faculty of Economics and Administrative Science, Selçuk University, Konya 42130, Türkiye; ecelep@selcuk.edu.tr
4   Department of Tourism Management, Manavgat Tourism Faculty, Akdeniz University, Antalya 07600, Türkiye; gates@akdeniz.edu.tr
5   Department of Social Work, Manavgat Social Sciences and Humanities Faculty, Akdeniz University, Antalya 07600, Türkiye; arifaytekin@akdeniz.edu.tr
*   Correspondence: aylinalan@akdeniz.edu.tr (A.A.A.); ozguryayla@akdeniz.edu.tr (O.Y.); Tel.: +90-5331-508-511 (A.A.A.); +90-5542-693-592 (O.Y.)

**Abstract:** The primary purpose of this research is to reveal the moderator role of self-congruity in the effect of consumer return on investment, aesthetics, service excellence, and playfulness as the sub-dimensions of destination experience value on destination brand equity and in the effect of destination brand value on behavioral intention. The research data were obtained utilizing the survey method. The findings of the study revealed that all dimensions under the destination experience value significantly and positively affect the destination brand value. When the effect of the overall brand value of the destination on the behavioral intention is evaluated, the brand value affects the behavioral intention positively and strongly. It was concluded in the study that self-congruity has a moderator effect on customer return on investment and playfulness dimensions on destination brand value, whereas the construct of self-congruity does not have a moderator effect on service excellence and aesthetic dimensions on destination brand value and that self-congruity has a moderator effect on destination brand value perception on behavioral intention. The study provides important practical contributions to both destination marketing managers and national and local authorities.

**Keywords:** destination experience value; destination brand value; behavioral intention; self-congruity





## 1. Introduction

The world is experiencing rapid economic and social changes. While the economy is changing in this direction, the economic value obtained from goods and services has also left its place for the value obtained through experience [1]. Therefore, understanding the processes underlying and driving consumer behavior has become an increasingly critical area of research. The fact that consumers have different perspectives also indicates that there are unique differences in the process. The demarcation line here is not quantitative attributes of the product but experiential aspects that consumers can easily associate with its value. It is this psychological aspect of consumer behavior that makes the difference in an individual's purchase of any product.

A destination encompasses both its physical location and its metaphysical significance, which is shaped by the interconnected web of meanings and values attributed to it [2] and can be perceived subjectively depending on the itineraries, cultural background, visiting

purpose, education level, and past experiences of consumers [3]. Since tourism takes place in destinations, people visit destinations with the intention of exploring attractions, engaging in recreational activities, and forming memorable vacation experiences that arise from their interactions in the places they travel to [4]. Regardless of tourism activity, destination choice is a critical issue [5]. Therefore, the dynamic interaction of all kinds of cognitive, behavioral, and environmental events that affect the preferences of tourists, who are the consumers of destinations, is an issue that should be considered.

In today's competitive environment, destinations need to differentiate themselves to be successful. One of the most effective methods in the formation of this differentiation process has been brand value studies. The word "brand" has various denotations. The American Marketing Association defines a brand as "A name, term, sign, symbol, design, or a combination of all these, which serves to identify, describe, and distinguish a product from its competitors" [6]. According to a different definition, which approaches the concept of "brand" in terms of experience, it is "all the tangible or intangible benefits obtained from a product; in short, it is the entire customer experience" [7]. "Brand value" refers to the extra value that is attributed to both the product and the consumer as a result of the favorable impressions generated by a powerful brand name in the consumer's perception. Moreover, the concept of "brand equity" is not only developed for goods and services but has also been used for tourism destinations, and destination research has suggested that the universality of a brand should be considered regarding the characteristics of tourism and destination [8]. The higher the brand value of a business or destination, the more likely consumers will prefer it, and this will result in higher profitability [9,10].

Self-congruity theory is one of the most frequently used theories in destination experience and destination brand equity studies [11–14]. In the broadest sense, the self is a complex, organized, and dynamic system of learned attitudes, beliefs, and value judgments that people have about themselves [15]. Self-congruity refers to the congruence between a consumer's self-concept and the personality of a brand that consumers perceive or encounter while establishing a relationship between themselves and the brand [16]. In the tourism literature, the concept of "self-congruity" emerges as the process of comparing the tourist's self-image with the destination. In this context, the more congruence tourists have between their self-image and the destination, the more likely they are to prefer the destination.

Tourism destinations strive to offer unforgettable experiences to their visitors to compete and are effective while competing. Building brand equity in a particular region or place as a tourist destination is a long and continuous process that depends on many factors [17]. One of the primary difficulties in establishing a unique destination brand lies in comprehending the entity of a place's identity and determining the fundamental qualities that shape its distinctive character [2]. The rationale underlying this is that consumers have a preference for the destinations that are based on their characteristics. Destination brand equity is composed of a blend of services developed and provided in cooperation with local stakeholders that contribute significantly to the experience quality [2].

According to the literature review, there are several studies on destination value and the relationship between brand value and brand identity. Some of these studies [18] argue that destination branding creates an attractive and unique brand identity. This brand identity is often used as a mediating variable in theoretical models regarding the brand behavior intention of consumers [19]. Some other studies [20] emphasize that destination brand identity is one of the strategies for sustainable destination management. In another study [21], the influence of destination brand identity, brand loyalty, and perceived benefits of tourism on the loyalty behavior of residents in destination branding was examined. Kumar and Kaushik [22] stated that various dimensions of destination brand experience had a number of impacts on destination brand identity, which subsequently influenced the trust and loyalty of tourists to a tourism destination. As mentioned in previous studies, brand identity has been examined in relation to destination value and behavioral intention from different perspectives. Therefore, the current study model was designed to approach

the field from a different perspective. This study aims to investigate how destination experience value affects brand equity and behavioral intention, and the moderator role of self-congruity in this regard, and brand identity has not been included in the model.

When considered in general, there are several studies regarding experience value [1,23–25], destination brand equity [10], the relation between behavioral intention and perceived value [26,27], service excellence strategy [28], self-congruity and destination preference behavior [11–13,29,30]. However, no empirical research has been conducted to determine the moderating role of self-congruity in the effect of experience value on brand equity and behavioral intention. In the literature, prominent researchers in this field, including Aaker [31], argue that, to date, there have been very few empirical studies supporting the validity of self-congruity theory [32]. In this context, it is thought that the study will contribute to filling the gap in the literature. It also aims to empirically verify the role of self-congruity in the effect of destination value on destination brand value and behavioral intentions of tourists visiting Side (Antalya, Turkey), which is an important destination in terms of both touristic attractions and carrying capacity.

## 2. Literature Review

### 2.1. Experiential Value, Brand Equity, and Behavioral Intention

Tourism is a sector that deals with the experience of tourists concerning visiting, learning, seeing, having fun, and living differently. In this context, all the things that tourists encounter within a location, whether they pertain to actions or perceptions, thoughts or feelings, explicit or indirect, are open to being felt or undergone [33]. Tourism is a sector that deals with the tourist's experience of visiting, seeing, learning, having fun, and living differently. In this sense, everything that tourists experience in a destination, whether behavioral or perceptual, cognitive or emotional, expressed or implied, can be experienced [33]. As Yu [34] states, experiential value is a cognitive evaluation and preference of the value personally experienced by the customer at the time of service delivery, and tourists shape their personal experiences using the available resources and facilities offered by the destination [35]. Since we live in an experience economy, many businesses today focus on creating valuable and memorable experiences for customers [36]. Therefore, the understanding of value creation has been changing and shifting from a product-oriented financial perspective to a personalized experience-oriented perspective [37]. Particularly in the tourism sector, whose products have different qualities from physical products, people are highly influenced by experiences and recommendations [38].

There are two dimensions of experiential value, as stated in the study conducted by Tsai and Wang [39]. The first one is "extrinsic-intrinsic value", and the other one is the "activity" dimension, which can be divided into active and reactive categories. According to this dimensioning, playfulness is the active source of intrinsic value, while aesthetics is the reactive of intrinsic value. While service excellence is the reactive source of extrinsic value, Consumer Return on Investment (CROI) is regarded as the active source of extrinsic value. The CROI, identified as the dynamic generator of external value, involves actively reserving financial, time-related, behavioral, and psychological assets that bring in potential gains, and the consumers can realize this yield in the form of economic advantages [25]. Aesthetic value, which is the reactive source of intrinsic value, is explained as the dimension of consumption that appeals to emotions. Aesthetic value, which is an element of experiential value, comes to the fore for tourist consumption in parallel with the increase in life quality. In the tourism sector, aesthetics is an important dimension in creating the attractiveness of the touristic product [40]. In addition, "aesthetic value" is an experiential value that emerges when the products that stand out with their aesthetic beauty are consumed [41]. As stated by da Costa Mendes et al. [35], tourists consume not only a reality but also representations and symbols of reality, which may be defined as "consumption aesthetics" if they stay at the destination. When the literature is examined, playfulness behavior, which is included in the active intrinsic value section, provides inner pleasure by focusing on events that drag the person to a point where they can escape from the demands of everyday

life [25]. Individuals with playfulness characteristics can transform any environment into a more entertaining environment [42]. Besides, the dimension of playfulness arouses concentration and curiosity in people [20]. As for service excellence, as the fourth and last dimension and on the external reactive side, it is not just about providing luxury-level service, but it means the ability of service providers to consistently meet and sometimes even surpass the anticipated outcomes of customers [43]. What is more, service excellence not only results in customer satisfaction but also provides broader customer loyalty and long-term profitability [28].

Brand attachment, which is of great importance in reflecting the emotional aspect of tourist–destination relationships [44], positively affects destination brand equity, connects tourists to the destination, and tourists with high brand equity gain an advantage in the competitive advantage of destinations through positive word-of-mouth marketing [45].

In their study, Zhang et al. [46] state that there is limited research on the antecedents and consequences of noteworthy tourism experiences. In their study, Zhang, Wu & Buhalis [47], state that there is limited research on the antecedents and outcomes of memorable tourism experiences. According to the same research result, the image of the country and destination is a factor that also affects the revisit intention with the mediator effect of the memorable tourism experience. Luo et al. [48] concluded that destination brand value affects destination loyalty of tourists and destination loyalty is also related to the intentions of tourists to travel and suggest that destination again. This situation shows that the destination brand value is effective on the behavioral intentions of tourists. According to the findings obtained from the study conducted by Lee et al. [49] on virtual golf simulator players, this perceived value significantly predicts behavioral intention. Altınay et al. [38] also state that consumers are influenced not only by tourism but also by experiences and shared ideas regarding all purchasing processes. Here, the importance of the concept of experiential value becomes known. However, since experiential value is also a new concept in marketing, it is important to design and implement empirical studies to validate this new concept and place it on a foundation that will enable it to evolve. It is prudent to mention that experiential value is usually analyzed under various dimensions in the literature. Batat [50] presented the new experiential marketing mix as 7E in his study. These are Experience, Exchange, Extension, Emphasis, Empathy, and Emotional touchpoints. In their study, Matwichk et al. [25] also mentioned two other main dimensions, Hedonic/Intrinsic Value and Utilitarian/Extrinsic Value, in the experiential value scale they developed and the hierarchical model of experiential value they proposed. In the study, the researchers stated that Hedonic/Intrinsic Value consists of two sub-dimensions, namely, aesthetics and playfulness, and the aesthetics sub-dimension consists of visual appeal and entertainment. The playfulness sub-dimension includes escapism and enjoyment. Besides, utilitarian/extrinsic value consists of service excellence and customer ROI, while customer ROI consists of efficiency and economic value. To the findings of [25] and considering these four elements: Aesthetics, Playfulness, Service Excellence, and Customer ROI the subsequent hypotheses have been put forth:

**H1** : *Experiential value has a significant effect on the brand value of the destination.*

**H1a** : *Return on customer investment has a significant effect on the brand value of the destination.*

**H1b** : *Service excellence has a significant effect on the brand value of the destination.*

**H1c** : *Aesthetics have a significant effect on the brand value of the destination.*

**H1d** : *Playfulness has a significant effect on the brand value of the destination.*

**H2** : *Destination brand value has a significant effect on behavioral intention.*

### 2.2. Self-Congruity

Tourists tend to travel for various tourism purposes when the ever-changing world conditions are taken into consideration. For this reason, destinations that can emphasize the

attractions that are compatible with tourists' selves will be successful [11,13,14,29,51–54]. Thus, the importance of "self-congruity theory" in the process of destination brand equity has been increasing.

When the "Self-congruity" literature, which is defined as the congruity between the personality types and the brand personality, was examined, it was seen that four different types of congruity were mentioned [55,56]: real self-congruity, ideal self-congruity, social self-congruity, and ideal social self-congruity. Self-congruity is based on the congruence between one's real self and perception of the product/brand. Ideal self-congruity is the congruence between a person's ideal self and his/her perception of the product/brand. The ideal self is accepted as the self-regarding the ideals and goals that the person believes or desires to be [57]. The social and ideal social self is included in the social personality of the individual [58]. Social self-congruity refers to the self-perception in the minds of others, while ideal social-self-congruity refers to the self-perception that you want others to think [59]. Consumers have a tendency to assess the brand by matching their real, ideal, and social self-concepts with the brand-user image [60].

Just as people possess an image of their own, they also have an image of products, suppliers, and services. Naturally, the research on self-concept theory has shown that consumers hold favorable opinions regarding brands and products that they perceive as consistent with their self-image [56,61,62]. Following the purchase and use of the product by the consumers, the product starts to bear some meanings to the consumer, and these meanings are evaluated by the self together with the self-image, and the consumer gets an idea about the product image. Following the ideas and observations acquired, the consumer starts to establish a connection between the self and the product [59]. The perception of product image and self-image that will develop in consumers will result in positive self-conformity that will lead to purchase motivation [12,63].

In consumer behavior research, the role of self-concept in determining consumer behavioral intention in different segments of products and services is well supported [64]. In accordance with the self-congruity theory, individual behavior is influenced by the product image (e.g., what kind of people use that product) and the consumer's self-concept [12,65]. Landon [66], in the study he conducted on self, ideal self, and consumer purchase intention, emphasized that it would not be efficient to evaluate the relationship between self-concept and purchase preference as a simple and direct relationship of congruity or incongruity.

Sirgy and Su [29] used the concept of self in their research on consumers to explain the psychological basis of travel behavior. In their study, they also created a conceptual model for self-congruity and travel behavior. In the model, it is stated that several qualifications and the atmosphere of the destination are closely associated with the image of the individuals visiting the destination. Subsequently, this image is assessed in relation to specific dimensions of the tourist's self-concept, aiming to gauge the extent of self-congruity. As a consequence, self-congruity is associated with travel behavior. On the other hand, according to the model, travel behavior is impacted not solely by self-congruity but also by various factors, such as functional congruity or the utilitarian characteristics of the destination, and the extent to which these characteristics match the expectations of visitors is also important. Segota et al. [12] stated that the self-concept has been extensively researched in marketing and tourism literature and that self-congruity has an effect on destination satisfaction.

As can be understood from the tourism literature, psychological and functional variables, such as self-concept and attitudes, positively affect the commitment of tourists to a destination [64]. The tangible characteristics of destinations fall under the functional category, while intangible features are categorized as symbolic [23]. Tourists expect to receive symbolic and functional benefits when creating a tourism experience [35]. In the study conducted by Zhu et al. [67], it was discovered that both utilitarian and symbolic brands not only attract the consumer's real self but also resonate with their ideal selves. As stated by Yuan and Wu [68], experiential marketing aims to enhance customer satisfaction

by offering symbolic and functional values derived from sensory perceptions, cognitive perceptions, and the quality of services.

Today, self-congruity has a significant role in customer behavior characteristics that affect brand satisfaction, product features, marketing effectiveness, brand loyalty, preference, and evaluation [65]. Kim et al. [16] allege that consumers will be able to express their values by using a brand with a personality compatible with themselves, and they will also go through a social adaptation process. Chon [51], in a study in which he integrated the theory of self-congruity into the tourism sector, revealed a positive interaction between self-congruity and tourist satisfaction. Based on the consequences, the tourists experiencing self-congruity reported greater satisfaction with the travel destination. Litvin and Goh [69] examined whether real and ideal self-congruity influences the satisfaction of tourists. The results of the study have confirmed the applicability of the self-image congruity concept for tourism research.

The study conducted by Beerli et al. [11] was designed to empirically verify the role of self-congruity in destination preference and the moderator variables affecting behavior. According to the results of the analysis, the higher the similarity between one's actual and ideal self-concepts and the destination's image, the greater the likelihood that the tourist will prefer that destination. This effect is also moderated by considerations, such as whether the individual has visited the destination before or is highly involved in vacation travel. The study concluded that the previous visit of the individual to the destination decreases the effect of self-congruity, while the high involvement of the individual in vacation travel increases the effect of self-congruity. In this regard, the other research hypotheses that are put forward because of the literature review are as follows:

**H3** : *Self-congruity has a moderator role in the effect of experiential value on destination brand value.*

**H3a** : *Self-congruity has a moderator role in the effect of customer return on investment dimension of experiential value on destination brand value.*

**H3b** : *Self-congruity has a moderator role in the effect of the service excellence dimension of experiential value on destination brand value.*

**H3c** : *Self-congruity has a moderator role in the effect of the aesthetic dimension of experiential value on destination brand value.*

**H3d** : *Self-congruity has a moderator role in the effect of the playfulness dimension of experiential value on destination brand value.*

**H4** : *Self-congruity has a moderator role in the effect of destination brand value perception on behavioral intention.*

## 3. Methodology

### 3.1. Research Instrument

The measurement scales employed to assess the variables established in the research model were formulated following an extensive review of the existing literature. In this context, fourteen items measuring the value of destination experience were used by adapting from the study of Tsai and Wang [39]. The four expressions chosen for destination brand value were adapted from the study of Im et al. [70]. For self-congruity, which was determined as the moderator variable of the research, a total of four expressions used by Kumar [71] were preferred. Then, to determine behavioral intentions, a 3-item scale was used by adapting from the study of Tsai and Wang [39]. Therefore, a set of 25 statements was administered to tourists who were present at the destination, utilizing a 5-point Likert scale (1 = strongly disagree; 5 = strongly agree).

### 3.2. Study Area

The research was conducted in Manavgat, which is 75 km away from Antalya and is one of Turkey's most important tourism destinations. Manavgat has a qualified supply

source and outstanding attractions (historical artifacts, natural beauties, sea, sand, climate, etc.) in terms of its current tourism potential. It is observed that more than four million tourists visited Manavgat when the 2022 data were evaluated [72]. The data reveal that one in every three tourists who travel to Antalya pays a visit to Manavgat. Considering these factors, Manavgat was selected as the population of this research.

### 3.3. Sampling and Data Collection

The research data were obtained from the tourists who visited Manavgat during the months of July, August, and September 2022, which are the busiest times of the Manavgat destination in terms of tourism activities. The researchers collected the data themselves using convenience sampling. Before progressing to the data collection phase of the research process, a pilot study was carried out to assess the research's validity, reliability, clarity, and overall comprehensibility. In this context, the questionnaire was applied to forty-one people on 5–11 July 2022. According to the feedback received from the participants who answered the pilot study questionnaire, no formal errors were detected. Besides, it was determined that the factor loads of the scale were a minimum of 0.54, and the Cronbach alpha values were a minimum of 0.83 for each construct. Then, all these values were accepted to be appropriate [73], and the stage of collecting the actual data for the research was started.

At the end of the three months, 525 questionnaires were obtained, the questionnaires with incorrect and missing data were excluded, and analyses were conducted through 518 questionnaires. Due to the high risk of common method bias in the research carried out in social sciences [74], response-enhancing techniques were applied, and for each questionnaire, a cover page was prepared to contain information such as "Participation is optional", "Any information collected during our research will be kept confidential", "There are no correct or incorrect answers in this questionnaire" [75]. Regarding statistical solutions, Harman's single-factor test was applied. After entering all variables into an exploratory factor analysis (EFA), the unrotated factor solution revealed that no single element accounted for the majority of the variance (the largest identified factor explained 25.7% of the variance).

### 3.4. Data Analysis

Initially, the research data were loaded into the SPSS program. The data screening process was applied before the determination of the relationships between the variables considered within the scope of the research. The process of data screening was conducted in three distinct stages. In this sense, an initial step involved data screening and the analysis of Mahalanobis distance to identify potential extreme values. As a consequence of this screening, 11 survey forms were excluded from the analysis due to the presence of extreme values (Mahalanobis' D (25) > 0.001). Subsequently, an assessment of multicollinearity was conducted. Following the analysis, it was ascertained that the VIF (Variance Inflation Factor) values remained below 5, and the tolerance values exceeded 0.10. Based on these results, it was determined that no issue of multicollinearity was present [73]. In the final phase, the skewness and kurtosis coefficients of the data were scrutinized. Skewness and kurtosis coefficients represent the extent of deviation from a regular distribution [76]. Both coefficients should ideally stay within the range of $\pm 2$ [76,77]. In the present study, skewness values for the relevant variables were found to range from $-0.911$ to $+0.212$, and kurtosis values varied from $-1.112$ to $+1.043$. Given these outcomes, data analysis was executed using AMOS software (IBM SPSS AMOS 22). Within the framework of AMOS program, the convergent and discriminant validity of each construct was assessed, followed by the calculation of coefficients for path analysis [78]. Additionally, to identify moderator effects, the Process macro (Hayes [79]; model 1) was employed.

## 4. Findings

### 4.1. Demographic Profile

The study's descriptive analysis reveals that 56% of the participants were female (n = 284), and 44% were male (n = 223). The majority of the participants (29.6%, n = 150) were between the ages of 26–35, and 24% (n = 122) were between 18 and 25. Then, respectively, comes 19.5% aged 36–45 (n = 99), 15.8% aged 46–55 (n = 80), 7.7% aged 55–65 (n = 39), and 3.4% aged 66 and over (n = 17). In addition, 52.9% of the participants were single (n = 268), and 47.1% were married (n = 239). A majority of participants held a bachelor's degree (198), followed by an associate degree (96) and a post-graduate degree (71). The number of secondary school graduates was 121, while the number of primary school graduates was 21. The examination of nationality indicated that 43.2% (n = 219) are either Russian or German tourists. This finding is not surprising at all considering that tourists of these nationalities visit the country the most [72].

### 4.2. Model Validity

Before examining the effect values between the variables in the developed model, the two-stage approach advocated by Anderson and Gerbing [78] was utilized. In this regard, confirmatory factor analysis, which is the first step, was applied, and the results are shown in Table 1. The study indicated an acceptable—good fit of the measurement model ($\chi^2$ = 969.608, df = 254, CMIN/df = 3.817, GFI = 0.906, NFI = 0.902, RMSEA = 0.078, CFI = 0.925, IFI = 0.926). The degree of $\chi^2$/df being below 5 indicates that the structure has an acceptable fit [80]. The fact that CFI, NFI, GFI, and IFI values are greater than 0.90 supports the good fit [81]. Because the RMSA value is below 0.80, the model fit is confirmed [82]. Therefore, the seven-factor structure, which is put forward theoretically in the measurement model, has been well-supported in the study.

**Table 1.** Description of the Participants (n = 507).

| Variables | Frequency | Percentage (%) |
|---|---|---|
| **Gender** | | |
| Male | 223 | 44.0 |
| Female | 284 | 56.0 |
| **Age** | | |
| 18–25 | 122 | 24.0 |
| 26–35 | 150 | 29.6 |
| 36–45 | 99 | 19.5 |
| 46–55 | 80 | 15.8 |
| 56–65 | 39 | 7.7 |
| 66 or more | 17 | 3.4 |
| **Marital Status** | | |
| Married | 239 | 47.1 |
| Single | 268 | 52.9 |
| **Education** | | |
| Primary | 21 | 4.1 |
| Secondary | 121 | 23.9 |
| Associate | 96 | 18.9 |
| Bachelor's | 198 | 39.1 |
| Post-graduate | 71 | 14.0 |
| **Nationality** | | |
| European | 219 | 43.2 |
| Asian | 288 | 56.8 |

Furthermore, alongside the data showcased in Table 2, the factor loadings of the items within each latent variable were incorporated. All factor loadings surpassed the threshold of 0.50, a value acknowledged in the existing literature [73]. Simultaneously, all computed t-values hold statistical significance at the $p \leq 0.001$ level. This implies that the items under the respective factors significantly contribute to measuring their corresponding constructs.

**Table 2.** SEM results of the research model.

| Factors/Items | Standard Loadings | t-Value | $R^2$ | CR | AVE | CA |
|---|---|---|---|---|---|---|
| Factor CROI: Consumer Return on Investment | | | | 0.922 | 0.790 | 0.937 |
| CROI1 | 0.897 | 26.902 * | 0.80 | | | |
| CROI2 | 0.901 | 27.166 * | 0.81 | | | |
| CROI3 | 0.893 | 26.902 * | 0.79 | | | |
| CROI4 | 0.864 | | 0.74 | | | |
| Factor SE: Service Excellence | | | | 0.892 | 0.735 | 0.893 |
| SE1 | 0.888 | 22.884 * | 0.78 | | | |
| SE2 | 0.854 | 21.757 * | 0.72 | | | |
| SE3 | 0.830 | | 0.68 | | | |
| Factor AES: Aesthetics | | | | 0.888 | 0.720 | 0.882 |
| AES1 | 0.837 | 19.855 * | 0.70 | | | |
| AES2 | 0.912 | 21.604 * | 0.83 | | | |
| AES3 | 0.793 | | 0.62 | | | |
| Factor PLYF: Playfulness | | | | 0.895 | 0.741 | 0.920 |
| PLYF1 | 0.828 | 22.401 * | 0.68 | | | |
| PLYF2 | 0.899 | 25.726 * | 0.80 | | | |
| PLYF3 | 0.864 | 24.069 * | 0.74 | | | |
| PLYF4 | 0.853 | | 0.72 | | | |
| Factor DBV: Overall Destination Brand Value | | | | 0.783 | 0.601 | 0.937 |
| DBV1 | 0.826 | 11.431 * | 0.68 | | | |
| DBV2 | 0.910 | 11.854 * | 0.82 | | | |
| DBV3 | 0.793 | 11.225 * | 0.63 | | | |
| DBV4 | 0.515 | | 0.26 | | | |
| Factor SC: Self-Congruity | | | | 0.803 | 0.621 | 0.907 |
| SC1 | 0.741 | | 0.54 | | | |
| SC2 | 0.919 | 19.213 * | 0.84 | | | |
| SC3 | 0.837 | 18.000 * | 0.70 | | | |
| SC4 | 0.626 | 13.237 * | 0.39 | | | |
| Factor BI: Behavioral Intention | | | | 0.903 | 0.757 | 0.889 |
| BI1 | 0.918 | | 0.80 | | | |
| BI2 | 0.912 | 28.907 * | 0.83 | | | |
| BI3 | 0.774 | 21.311 * | 0.59 | | | |

* $p < 0.001$.

The alpha values for each construct of the scales were determined to vary between 0.882 and 0.937. This result shows that the scale is dependable [83]. In addition, the lowest composite reliability (CR) value of 0.783 indicates that the structure reliability is provided [84]. Finally, the average variance extracted (AVE) values were scrutinized concerning convergent validity. All values were 0.50 and above, thus proving convergent validity [85].

Table 3 displays the model's discriminant validity assessment. The findings indicate that the AVE for each construct surpasses the corresponding values in the related row. Regarding the confirmation of discriminant validity, the Maximum Shared Variance (MSV) and Average Shared Variance (ASV) values are all below the extracted AVE values [73]. Consequently, the construct also demonstrates discriminant validity [86].

**Table 3.** The results of discriminant validity.

| Factor | MSV | ASV | 1 | 2 | 3 | 4 | 5 | 6 | 7 |
|---|---|---|---|---|---|---|---|---|---|
| 1. CROI | 0.502 | 0.318 | 0.888 [a] | | | | | | |
| 2. SE | 0.538 | 0.365 | 0.528 | 0.857 [a] | | | | | |
| 3. AES | 0.494 | 0.344 | 0.538 | 0.561 | 0.848 [a] | | | | |
| 4. PLYF | 0.565 | 0.344 | 0.464 | 0.534 | 0.521 | 0.860 [a] | | | |
| 5. DBV | 0.404 | 0.404 | 0.550 | 0.544 | 0.538 | 0.687 | 0.775 [a] | | |
| 6. SC | 0.128 | 0.169 | 0.292 | 0.324 | 0.298 | 0.354 | 0.451 | 0.788 [a] | |
| 7. BI | 0.538 | 0.407 | 0.663 | 0.650 | 0.651 | 0.585 | 0.536 | 0.301 | 0.870 [a] |

Note: [a]: square root of AVE.

### 4.3. Hypothesis Tests

In the second stage of the research, to evaluate the hypotheses that were determined depending on the research purpose, the path analysis proposed by Anderson and Gerbing [78] was used. Upon the examination of structural model fit values, it is clear that the data are within acceptable limits ($\chi^2$ = 969.608, df = 254, $\chi^2$ = 3.529, GFI = 0.901, NFI = 0.927, RMSEA = 0.074, CFI = 0.946, IFI = 0.947). It was revealed that Consumer Return on Investment has a significant effect on the perceived brand value of the destination ($\beta$ = 0.26, t = 7.155, $p < 0.001$). Service Excellence has positively affected the perceived brand value of the destination ($\beta$ = 0.22, t = 5.860, $p < 0.001$). Similarly, a positive change in the Aesthetics dimension increases the destination brand value ($\beta$ = 0.20, t = 5.538, $p < 0.001$). Playfulness, which is a sub-dimension of destination experience value, is found to have a positive effect on destination brand value ($\beta$ = 0.29, t = 7.551, $p < 0.001$). As a result, all dimensions within the context of destination experience value have a significant positive effect on destination brand value. Considering these results, $H_{1a}$, $H_{1b}$, $H_{1c}$, and $H_{1d}$ are supported. Additionally, when the effect of the overall destination brand value on the behavioral intention is examined, the value has positively and significantly affected intention ($\beta$ = 0.71, t = 10.041, $p < 0.001$). This outcome substantiates the validity of $H_2$.

### 4.4. Moderator Effect

In order to assess the potential moderating function of self-congruity in the relationship involving Consumer Return on Investment, Service Excellence, Aesthetics, and Playfulness—sub-dimensions of experiential value—along with destination brand value, and their influence on behavioral intention, a regression analysis employing the bootstrap technique was carried out. According to Hayes [79], the Bootstrap method offers more dependable outcomes compared to the conventional Baron and Kenny [87] approach. Analyzes were performed utilizing Model 1 within the Process Macro framework developed by Hayes [79]. For analysis, the bootstrap technique was employed with 5000 resampling iterations. The results of the moderator effect are displayed in Table 4.

The findings demonstrate that the impact of Consumer Return on Investment, self-congruity, and their interaction on destination brand value perception, the outcome variable, holds significance. The noteworthy value of the interactional effect variable, indicating the potential presence of moderation, underscores that self-congruity exerts a moderating influence (=−0.10, 95% CI [0.040, 0.166], $p < 0.001$). Therefore, $H_{3a}$ is supported. The estimation variables incorporated in the regression analysis explain approximately 42% of the variation in destination brand equity. The self-congruity regarding the destination increases, and the effect of Consumer Return on Investment on the destination brand value perception also changes positively. In this sense, the influence power is 27% when the self-congruity level is low. The power level is 38% at the medium level, and 48% at the high at self-congruity level. In Playfulness, which is one of the destination experiential value dimensions, the moderator effect of self-congruity is on the effect of destination brand value. (=−0.09, 95% CI [0.033, 0.148], $p < 0.05$). According to these results, $H_{3d}$ has been supported. Within the scope of the details concerning the moderator effect, it is shown that

as self-congruity increases, the power between these variables increases as well. Because, in that interaction, the influence power is determined to be 46%, in cases where self-congruity is low, it is 55% at the medium level and 64% at a high self-congruity level. On the other hand, the dimensions of Service Excellence and Aesthetics, which are the sub-dimensions of the destination experiential value, as well as the self-congruity and the interactional term, had no statistically significant effect on the perception of destination brand value, which is the outcome variable ($p > 0.05$). On the basis of these results, $H_{3b}$ and $H_{3c}$ are not supported.

**Table 4.** Moderated Effect Result.

| Moderating Effect: | | | | | Overall Destination Brand Value | | |
|---|---|---|---|---|---|---|---|
| | | | | | $\beta$ | Confidence Interval | |
| Hypothesis 3a | | | | | | Min. | Max. |
| Cons. Return on Investment (X) | | | | | 0.77 * | 0.533 | 1.009 |
| Self-congruity (W) | | | | | 0.70 * | 0.458 | 0.951 |
| X.W (Interaction) | | | | | 0.10 * | 0.040 | 0.166 |
| $R^2$ | | | | | 0.40 | | |
| Self-congruity | $\beta$ | S.E. | t | LLCI | ULCI | | |
| Low | 0.27 * | 0.04 | 5.80 | 0.184 | 0.384 | | |
| Middle | 0.38 * | 0.03 | 11.80 | 0.319 | 0.446 | | |
| High | 0.48 * | 0.04 | 11.30 | 0.402 | 0.571 | | |
| | | | | | Overall Destination Brand Value | | |
| | | | | | $\beta$ | Confidence Interval | |
| Hypothesis 3b | | | | | | Min. | Max. |
| Service Excellence (X) | | | | | 0.54 * | 0.285 | 0.788 |
| Self-congruity (W) | | | | | 0.44 ** | 0.150 | 0.728 |
| X.W (Interaction) | | | | | −0.03 NS | −0.101 | 0.036 |
| $R^2$ | | | | | 0.42 | | |
| | | | | | Overall Destination Brand Value | | |
| | | | | | $\beta$ | Confidence Interval | |
| Hypothesis 3c | | | | | | Min. | Max. |
| Aesthetics (X) | | | | | 0.51 * | 0.277 | 0.744 |
| Self-congruity (W) | | | | | 0.42 * | 0.160 | 0.683 |
| X.W (Interaction) | | | | | −0.02 NS | −0.090 | 0.039 |
| $R^2$ | | | | | 0.39 | | |
| | | | | | Overall Destination Brand Value | | |
| | | | | | $\beta$ | Confidence Interval | |
| Hypothesis 3d | | | | | | Min. | Max. |
| Playfulness (X) | | | | | 0.79 * | 0.675 | 1.105 |
| Self-congruity (W) | | | | | 0.58 * | 0.358 | 0.819 |
| X.W (Interaction) | | | | | 0.09 ** | 0.033 | 0.148 |
| $R^2$ | | | | | 0.53 | | |
| Self-congruity | $\beta$ | S.E. | t | LLCI | ULCI | | |
| Low | 0.46 * | 0.05 | 9.81 | 0.366 | 0.549 | | |
| Middle | 0.55 * | 0.03 | 17.16 | 0.485 | 0.612 | | |
| High | 0.64 * | 0.04 | 16.02 | 0.561 | 0.718 | | |
| | | | | | Behavioral Intention | | |
| | | | | | $\beta$ | Confidence Interval | |
| Hypothesis 4 | | | | | | Min. | Max. |
| Overall Destination Brand Value (X) | | | | | 0.84 * | 0.651 | 1.277 |
| Self-congruity (W) | | | | | 0.52 * | 0.176 | 0.864 |
| X.W (Interaction) | | | | | 0.12 ** | 0.027 | 0.195 |
| $R^2$ | | | | | 0.30 | | |
| Self-congruity | $\beta$ | S.E. | t | LLCI | ULCI | | |
| Low | 0.43 * | 0.07 | 6.00 | 0.292 | 0.576 | | |
| Middle | 0.54 * | 0.05 | 10.82 | 0.447 | 0.645 | | |
| High | 0.65 * | 0.05 | 11.05 | 0.540 | 0.774 | | |

\* $p < 0.001$; \*\* $p < 0.05$; NS: no significant.

The impact on behavioral intention, as an outcome variable, is significantly influenced by destination brand value perception, self-congruity, and their interaction. Notably, the significant value of the interactional effect variable, indicating the presence of a moderator effect, signifies that self-congruity moderates the connection between these variables (=−0.12, 95% CI [0.027, 0.195], $p < 0.05$). Therefore, H4 is supported. Upon analyzing the details of the moderating effect, it can be defined that as the self-congruity regarding the destination increases, the effect of the destination brand value perception on the behavioral intention also changes positively. In this sense, in cases where self-congruity is low, the influence power is determined to be 43%, it is 54% at a medium level, and 65% at a high self-congruity level.

## 5. Discussion

This study has examined the effect of destination experience value on brand equity and the moderator role of self-congruity in the effect of destination brand equity on behavioral intention. The findings reveal a positive relationship between destination experience value and brand equity, in line with previous studies in the literature. In addition, it has been found that destination brand equity has a positive effect on the behavioral intention of visitors.

These findings highlight the importance of successful marketing and management of touristic destinations. They can enhance brand equity by providing visitors with unique and satisfying experiences. For instance, natural beauty, cultural richness, or entertainment options of a destination can offer unforgettable experiences to visitors and improve brand equity [88]. These experiences shape the positive perception of visitors about the destination and influence their behavioral intentions.

Destination brand equity is a process that goes beyond forming the perception of visitors. It is a process in which this perception needs to be maintained and strengthened. In this regard, destination management and marketing play an important role. Well-designed marketing strategies ensure that the destination leaves a positive impression on its target masses and can increase behavioral intention. In this context, tools, such as digital marketing, social media strategies, and effective content management, can be instrumental in increasing destination brand equity.

The relationship between destination experience value and brand equity indicates that tourist destinations are based not only on their tangible characteristics but also on the experiences offered to visitors [89]. The findings of this study emphasize that experience value is an important factor in enhancing destination brand equity.

Particularly, the self-congruity factor moderates the relationship between tourist destination experience and brand equity. In other words, a visitor's own personality qualities and values can shape the relationship between experience value and brand equity. For example, the link between experience value and brand equity may be stronger for a visitor if a natural and tranquil destination is more in congruence with their intrinsic values.

As for behavioral intention, it reflects visitors' intention to visit a destination or purchase its services. The effect of brand equity on consumer behavioral intentions was also examined in previous research [90,91]. The findings of this study have suggested that increasing destination brand equity can positively influence the behavioral intention of visitors. However, the moderating effect of self-congruity on this relationship is important in responding to the specific needs of each visitor type and segment. Tourism destinations can increase behavioral intention by not only offering travel experiences but also by providing services that are tailored to the needs of the target groups.

These findings provide an in-depth insight into the brand management strategies of touristic destinations. They can increase brand equity by providing experiences that are congruent with visitors' own identities and values. Especially understanding the target groups and assessing their self-congruity is important in shaping the marketing and communication strategies of a destination.

As a consequence, this research enlightens the complex and reciprocal interaction between experience value, brand equity, behavioral intention, and self-congruity of tourist destinations. The findings of the study could serve as a basis for understanding how destinations can optimize experience value and brand equity to achieve sustainable competitive advantage and create visitor loyalty. Thus, future studies are important to examine this relationship more in-depth and understand how it changes under different destination types or cultural contexts. Moreover, it is necessary to delve deeper into how different marketing strategies and segmentation approaches can be optimized.

## 6. Conclusions

In terms of providing a competitive advantage between destinations, the destination experiences of the tourists and the destination brand value are at least as important as the touristic attractiveness and service quality offered by the destinations. Side as a town and its surroundings are considered a touristic destination that has integrated features with world-class facilities and unique beauty. In this study, considering these points, to contribute to Side's activities as a destination, the moderator role of self-congruity in determining the effect of the experience value of the tourists coming to the destination from the destination, the overall brand value of the destination and their behavioral intentions towards the destination were investigated. In line with the data obtained, a model was developed and evaluated.

The tourist experience is significant for a destination to become a brand. It is shown in the study that all the sub-dimensions of destination experience value (Consumer Return on Investment, Service Excellence, Aesthetics, and Playfulness) significantly and positively affect the brand value of the destination, unlike Tsai and Wang [39]. In their study, Tsai and Wang [39] state that CROI is the only experiential value dimension that significantly affects food brand image and destination food brand image. It is also stated that a unique destination image may be the antecedent of tourists' behavioral intention. Martins et al. [92] also concluded that the destination brand experience affects the behavioral intentions of the visitors.

The study's findings lead to the conclusion that self-congruity has a moderator effect on the effect of CROI and Playfulness, which are accepted as the sub-dimensions of destination experience value, on the overall destination brand value. The overall brand equity concerning the destination also has a positive and strong effect on the tourists' behavioral intentions. This finding indicates a similar result to numerous studies in the literature [48,49,92]. One of the results obtained is that self-congruity has a moderating role in the effect of destination brand value perception on behavioral intention. It is possible to see comparable results in previous studies [12,29,65,93]. According to the results of the study [30], which was conducted to reveal the effect of self-congruity on behavioral intention, it was stated that self-congruity positively affected the behavioral intention of tourists regarding the destination and that self-congruity had a partial mediating role between destination personality and behavioral intention. In the study of Kastenholz [52], it was concluded that self-congruity positively influenced the revisit intention.

In addition to these, the findings suggest that self-congruity does not play a moderating role in the impact of service excellence and aesthetic dimensions on destination brand value. This situation can be interpreted as the effect of service excellence and aesthetics on the overall destination brand value in the formation of brand value perceptions of the tourists participating in the research. However, there is no moderating role of self-congruity that was established between the destination and the tourists themselves.

In the tourism industry, the more important it is to make a country a brand, the more important it is to create the brand value of the cities that make up that country. Today, there may be cities whose brand values are more than the brand values of the countries they are in. To provide a competitive advantage in the context of the city, it is necessary to conduct several studies on tourism and promotional aspects. Presenting an attitude toward a behavior is one of the most important determinants in predicting and explaining

behavioral intention [94]. The results obtained from the study of Haji et al. [95] show that the quality of experience regarding the destination has a strong effect on behavioral intention and perceived value. In another study conducted on the subject [96], the findings unveiled that there is a notable correlation between the experience quality of the destination and behavioral intention. The results of the study by Hosany and Martin [54] emphasize that self-congruity, which includes real and ideal self-concepts, significantly affects the experience of tourists. Khazaei Pool et al. [97] state that the behavioral intention of the tourist is affected by the perceived experience and satisfaction, and the congruity between the self-concept and the destination personality also affects the perceived experiences of the tourists. The results of the current study have shown that the behavioral intentions of tourists concerning the destination are affected by the destination experience and overall brand values. In this effect, the congruity of the tourists' selves with destination characteristics is also prominent.

*6.1. Theoretical Implications*

The study provides various theoretical contributions to the existing literature via its findings. In the studies conducted in the literature [98,99], the importance of consumer self-congruity with destination in destination competition has been accepted. However, the effect of the destination experience on the overall brand value and the behavioral intentions of the tourists and the effect of self-congruity on this effect have been ignored. It is thought that the results of this study will contribute to future empirical studies in the field of tourism, particularly in terms of "self-congruity theory". As mentioned in the conceptual framework section of the study, self-congruity includes four different dimensions of self-congruity. In this study, self-congruity questions have been kept limited, and the detailed inclusion of all types of self-congruity in future research models will further our knowledge concerning brand equity.

The study has shown that destination experience value profoundly affects destination brand equity in all its sub-dimensions. Determining this effect in all sub-dimensions will provide significant additions to the already established knowledge in the literature. Another finding, the moderator role of self-congruity in the effect of destination experience value on overall destination brand equity, is thought to bring a distinct perspective to the destination marketing literature. Accordingly, an increase in self-congruity between the destination and the individual will increase the effect of destination experience value on overall destination brand equity. Apart from these contributions, the fact that there is no other study that examines the effects of the variables that make up the research model on each other based on destination marketing shows that the tested model will provide added value to the literature. The measurement model parameters of all variables (destination experience value, brand value, behavioral intention, and self-congruity) were found to be appropriate. According to these results, it is confirmed that all scales are valid instruments.

*6.2. Practical Implications*

The first of the practical contributions of this study is for destination marketers. At this point, the thing that will give destinations a competitive advantage is to understand the tourists' behavioral intention antecedents. Considering the wishes and expectations of the tourists, several steps should be taken to ensure congruence between the tourist and the destination. In this sense, it becomes important to understand the symbolic functions of the destination to grasp the complex structure of tourists' behavior. When the results of the study are examined, it can be observed that tourists who can harmonize their destination perceptions with their self-concepts are more likely to develop positive behaviors concerning the destination. To ensure that the services in the business have symbolic meanings for the tourists, several special reminders for tourists can be positioned within that business. Therefore, a connection can be provided between the tourist and both the business and the destination. In the enrichment of these symbolic meanings, in addition to the cooperation of both travel and accommodation businesses operating in Side,

non-governmental organizations can also contribute. Another practical contribution will be to central and local authorities. Considering Side to be an independent service production center as a tourism destination contributes to what should be focused on to increase local efficiency and increase the destination brand value. Considering the current marketing strategies of the country, the institutions or organizations that influence the practices to be developed or implemented for Side as a destination should not ignore the self-congruity of tourists in the effect of the destination experience value on brand value. With regards to this perspective, while creating the perception that tourists can relax and have a pleasant experience in the destination of Side, the personal characteristics of tourists should not be ignored.

It is suggested that the high level of moderator effect of self-congruity on the positive impact of destination brand equity on tourist behavior indicates that some activities should be carried out targeting the perceptions of tourists. In this context, using the power of the media effectively can be a good way to manage their perception. It is recommended to provide messages that will strengthen the destination brand image and ensure self-congruity in the tourist–destination relationship through visual/print media and web pages. In this way, the positive effect on brand value will be enhanced through the messages given, and this, in turn, will also have a significant positive impact on behavioral intention. The primary step for destination managers should be to encourage the use of the marketing communication strategy designed for destination brand value in all domestic and international promotional activities with a common message language. Moreover, increasing financial and moral support to the regions with good branding potential will enable destination managers and sector representatives to analyze tourism activities on both group and individual basis and take action.

*6.3. Limitations and Future Research*

In the study process, since it would be difficult to reach all the foreign tourists visiting Side due to restrictions such as transportation, time, and cost, the research data were collected from the population within the specified date range. In future studies, it will be possible to obtain more detailed and clearer results if surveys are applied, considering not only the domestic but the foreign tourists who come to a destination based on the entire year.

In the future, similar subjects can be studied with qualitative and quantitative methods in different destinations to attract attention to regional differences and similarities and so comparative studies can be conducted. What is more, the contribution of such studies could be increased in the literature by investigating the relationship between various variables other than the destination experience value and the overall destination brand value, such as brand attachment, brand loyalty, brand image, brand identity, and the moderator effect of other factors other than self-congruity on these variables in the formation of behavioral intention regarding a destination.

**Author Contributions:** This research paper has been agreed upon by all of the authors and carried out collaboratively, but each one of the authors has made individual contributions to the paper. H.K. provided project management and language supervision. O.Y. performed the analyses and focused on the process of testing the hypotheses and the scales of the article. A.A.A. and G.S.E. conducted an extensive literature review, contributed to the original draft, and formulated the research hypotheses. A.A. and E.G., contributed to the completion of the implications, limitations, and conclusion chapters. E.C. contributed to the completion of the methodology and discussion chapters. In addition, the authors used an internal audit system during the preparation phase and monitored each other for any potential setbacks. All authors have read and agreed to the published version of the manuscript.

**Funding:** This research received no external funding.

**Institutional Review Board Statement:** Not applicable.

**Informed Consent Statement:** Informed consent was obtained from all participants involved in the research.

**Data Availability Statement:** The data analyzed during this study are available on request from the corresponding author.

**Conflicts of Interest:** The authors declare no conflict of interest.

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
