# Peer review of "How Does Destination Experience Value Affect Brand Value and Behavioral Intention? The Moderator Role of Self Congruity"

_sustainability, doi:10.3390/su151814004_

Round 1
Reviewer 1 Report
The paper has a clear contribution both to theory and to practitioners. The research question and hypotheses are well grounded in the extant literature and the method is perfectly adequate.
Only one question: In the methodology, namely, in the "3.2 Sampling and data collection" section, you say "The researchers themselves collected the data using the convenience sampling method, adopting the principle of randomness." . Was it a convenience or a random sampling. That is, did you employ any mechanism to guarantee sample randomness (e.g., appraoching every third person that passes through a door). If you just tried to approach any tourist indiscriminately, you should just not say you adopted the principle of randomness.

There are some typos, repeated words, etc., as well some sentences that sound a bit odd. Please, find attached the manuscript with some minor suggestions.
Author Response
Thank you so much for your positive comments and assessment of our topic. Your comments and suggestions have been very helpful for further improving of our paper.
We greatly appreciate your feedback.
The changes which were made about the manuscript have been highlighted in our text.

Reviewer 2 Report
Dear authors,
I very much enjoyed reading and reviewing your paper.
The authors' manuscript entitled "How Does Destination Experience Value Affect Brand Value and Behavioral Intention? The Moderator Role of Self Congruity " is well crafted.
The paper is excellent in all its dimensions: bibliography review, hypotheses, methodological design, fieldwork, analysis and conclusions.
But, there is no discussion section. Please, compare your research with similar research so you can be drawn adequate conclusions. In the discussion section, some associated literature must be added to compare and contrast the key findings with the existing studies.
Author Response

(The authors gave the same response as above.)

Reviewer 3 Report
The topic of the article is interesting and has great practical relevance. The article is written in a very accessible manner for the reader. It does not contain any substantive formal errors. The assumptions of the study, the description of the research method and the in-depth analysis of the results are not objectionable. The conclusions are a multi-faceted commentary on the results obtained. I believe that after editorial revision, the article is ready for publication.
Author Response

(The authors gave the same response as above.)

Reviewer 4 Report
Thank you for submitting your manuscript to Sustainability for publication consideration.
Here are some items needing to be addressed that are intended to further improve the quality of the manuscript.
1. Sampling and Data Collection: Please explain the sampling process used regarding the "convenience sampling method, adopting the principles of randomness". The data collection method used to obtain responses should be mutually exclusive. The respondents were either selected at random, or they were accepted at convenience. It cannot be both.
2. Demographic Profile: Please provide a Table that presents this information in an organized fashion. You do not need to provide every single detail of the findings within the narrative, as this is the purpose of developing tables.
3. Please identify what tests were conducted to determine if common method variance was an issue. It is suggested that a Harmon's single factor test (Podsakoff et al., 2003), in conjunction with common latent factor ( CLF) analysis (Sreejesh, et al., 2019; Chakraborty, 2019), are needed.
4. Please elaborate on 'brand identification', and why it was not included within the study model.
5. Practical Implications read as being very general and the study's implications have already been used by tourist destinations for decades. Please clarify what the true significance of this study findings have upon any destination and how can their DMO's benefit from these findings? Perhaps developing some scenarios would assist in this clarification.
Author Response

(The authors gave the same response as above.)
